# Dual Effect of Glucuronidation of a Pyrogallol-Type Phytophenol Antioxidant: A Comparison between Scutellarein and Scutellarin

**DOI:** 10.3390/molecules23123225

**Published:** 2018-12-06

**Authors:** Qianru Liu, Xican Li, Xiaojian Ouyang, Dongfeng Chen

**Affiliations:** 1School of Chinese Herbal Medicine, Guangzhou University of Chinese Medicine, Guangzhou 510006, China; liuqianru2333@163.com (Q.L.); oyxiaojian55@163.com (X.O.); 2Innovative Research & Development Laboratory of TCM, Guangzhou University of Chinese Medicine, Guangzhou 510006, China; 3School of Basic Medical Science, Guangzhou University of Chinese Medicine, Guangzhou 510006, China; 4The Research Center of Basic Integrative Medicine, Guangzhou University of Chinese Medicine, Guangzhou 510006, China

**Keywords:** scutellarein, scutellarin, pyrogallol-type phytophenol, structure-activity relationship, glucuronidation, antioxidant

## Abstract

To explore whether and how glucuronidation affects pyrogallol-type phytophenols, scutellarein and scutellarin (scutellarein-7-*O*-glucuronide) were comparatively investigated using a set of antioxidant analyses, including spectrophotometric analysis, UV-vis spectra analysis, and ultra-performance liquid chromatography coupled with electrospray ionization-quadrupole time-of-flight tandem mass spectrometry (UPLC-ESI-Q-TOF-MS/MS) analysis. In spectrophotometric analyses of the scavenging of 1,1-diphenyl-2-picrylhydrazyl (DPPH^•^), 2,2′-azino-bis (3-ethylbenzothiazoline-6-sulfonic acid) (ABTS^+•^), and 2-phenyl-4,4,5,5-tetramethylimidazoline-1-oxyl 3-oxide radicals (PTIO^•^) and the reduction of Cu^2+^ ions, scutellarein showed lower IC_50_ values than scutellarin. However, in ^•^O_2_^−^-scavenging spectrophotometric analysis, scutellarein showed higher IC_50_ value than scutellarin. The analysis of UV-Vis spectra obtained after the Fe^2+^-chelating reaction of scutellarin showed a typical UV-Vis peak (λ_max_ = 611 nm), while scutellarein showed no typical peak. In UPLC-ESI-Q-TOF-MS/MS analysis, mixing of scutellarein with DPPH^•^ yielded MS peaks (*m*/*z* 678, 632, 615, 450, 420, 381, 329, 300, 288, 227, 196, 182, 161, and 117) corresponding to the scutellarein-DPPH adduct and an MS peak (*m*/*z* 570) corresponding to the scutellarein-scutellarein dimer. Scutellarin, however, generated no MS peak. On the basis of these findings, it can be concluded that glucuronidation of pyrogallol-type phytophenol antioxidants has a dual effect. On the one hand, glucuronidation can decrease the antioxidant potentials (except for ^•^O_2_^−^ scavenging) and further lower the possibility of radical adduct formation (RAF), while on the other hand, it can enhance the ^•^O_2_^−^-scavenging and Fe^2+^-chelating potentials.

## 1. Introduction

Phytophenol refers to a natural product of plant origin, bearing phenolic -OH. When three phenolic -OHs array at three adjacent positions (i.e., 1,2,3-positions) in a phytophenol molecule, the resulting compound is called a pyrogallol-type phytophenol. For example, scutellarein, baicalein, gallic acid, epithetaflagallin 3-*O*-gallate [1], and chalcone-tannin hybrid [2] are all pyrogallol-type phytophenols. The presence of phenolic -OH makes phytophenol (especially pyrogallol-type phytophenols) an important natural antioxidant. Some of the pyrogallol-type phytophenols may undergo glucuronidation.

Essentially, glucuronidation is a type of glycosidation reaction. During glucuronidation, an alduronic acid (e.g., β-d-glucopyranuronic acid) can condense with another molecule (such as aglycone) at the hemiacetal -OH to generate a glucuronide product. A recent report inappropriately named this product “glucuronate” [3]. Conventionally, the so-called “glucuronate” is a product obtained by condensation at carboxyl -OH (not hemiacetal -OH).

During plant and human metabolism, glucuronidation is usually catalyzed by various enzymes (e.g., uridine 5′-diphospho-glucuronosyltransferase [4,5,6]). This can explain why baicalein and its glucuronidated product baicalin (also called baicalein-7-*O*-glucuronide) co-exist in the same plant [7] and why baicalin can be found in serum after administration of oral baicalein [8].

As soon as the pyrogallol-type phytophenol is glucuronidated, its bioactivity changes to some degree. For example, scutellarein and its glucuronidated product scutellarin have been reported to display a difference in their neuroprotective effects [9,10,11]. Similar differences in bioactivity can also be observed in the case of baicalein and baicalin [12,13,14]. These differences can be attributed to 7-*O*-glucuronidation [11]. The relationship between these neuroprotective effects and antioxidant activity indicates that 7-*O*-glucuronidation may also alter the antioxidant activities of pyrogallol-type phytophenols. Nevertheless, until now, no experimental evidence has shown whether and how glucuronidation affects the antioxidant activity of pyrogallol-type phytophenols.

In the present study, we selected scutellarin and scutellarein as a pair of reference molecules to probe the possible effect of glucuronidation on pyrogallol-type phytophenol antioxidants. As seen in Figure 1, scutellarein is a pyrogallol-type (5,6,7-trihydroxy) phytophenol, which on glucuronidation of its 7-OH group, is transformed into scutellarein-7-*O*-glucuronide, scutellarin. If scutellarein and scutellarin exhibit a difference in their antioxidant activities, it can only be attributed to the glucuronidation of 7-OH. Their comparison was carried out using a set of antioxidant assays with specific mechanistic features (see below).

Although the comparison of the antioxidant activities was based only on a pair of phytophenols (scutellarein and scutellarin), our results can also provide important information for other pyrogallol-type phytophenol antioxidants (e.g., baicalein and baicalin) and even non-pyrogallol-type phytophenol antioxidants (e.g., quercetin and quercetin-3-*O*-β-d-glucuronide [15,16]). In addition, this comparative study may help to understand the chemical role of glucuronidation in human metabolism as well as in some diseases [4,17,18,19,20,21,22].

## 2. Results and Discussion

Antioxidant chemistry involves a reactive oxygen species (ROS)-scavenging reaction. Through ROS-scavenging, oxidative stress in cells can be effectively eliminated. To explore the antioxidant chemistry of two reference compounds, scutellarein and scutellarin, a set of spectrophotometric analyses was carried out for the scavenging of DPPH^•^, ABTS^+•^, PTIO^•^, and ^•^O_2_^−^ radicals and the reduction of Cu^2+^ ions. These spectrophotometric analyses revealed individual antioxidant characteristics. Spectrophotometric analysis of DPPH^•^ scavenging involves hydrogen atom transfer (HAT)-preferred multiple antioxidant pathways [23,24,25], while that of ABTS^+•^ scavenging involves electron transfer (ET)-preferred multiple antioxidant pathways [26,27]. Spectrophotometric analysis of PTIO^•^ scavenging involves ET plus proton transfer (PT) pathways in aqueous solution (pH 7.4) [28], while that of cupric-reducing antioxidant capacity (CUPRAC) is chiefly mediated by ET in aqueous solution (pH 7.4) [29,30]. The spectrophotometric analysis of ^•^O_2_^−^ scavenging, however, is based on the principle of pyrogallol autoxidation [31].

In the first four spectrophotometric analyses, scutellarein always gave lower IC_50_ values than scutellarin (Table 1). This quantitative comparison revealed that scutellarein possesses higher antioxidant capacity than scutellarin. The superiority of scutellarein was further verified by ultra-performance liquid chromatography coupled with electrospray ionization quadrupole time-of-flight tandem mass spectrometry (UPLC-ESI-Q-TOF-MS/MS) analysis.

As seen in Figure 2D, the product from the reaction of scutellarein with DPPH^•^ resulted in at least two chromatographic peaks (peak 1 and peak 2) of high intensity (about 3 × 10^5^). Both peak 1 and peak 2 could produce MS spectra with *m*/*z* 678. This value is exactly two units less than the sum of the molecular weights of scutellarein (M.W. 286) and DPPH^•^ (M.W. 394). Thus, the two peaks were preliminarily proposed to belong to the scutellarein-DPPH adduct. Peak 1, however, could be further broken in MS/MS spectra, giving rise to several fragments (*m*/*z* 632, 615, 450, 420, 381, 329, 300, 288, 227, 196, 182, 161, and 117, Figure 2F). Of these, *m*/*z* 227 and 196 were presumed to be from the DPPH^•^ moiety (Figure 3A) [32]. On the basis of these data (Figure 3A) and previous literature [33,34,35,36], the scutellarein-DPPH adduct (C_33_H_21_N_5_O_12_) was assumed to be formed via covalent bonding at the 7-*O* position of scutellarein (Figure 3B). Besides, in the reaction product of scutellarein with DPPH^•^, one weak chromatographic peak with *m*/*z* 570 value was found (about 2 × 10^3^, Figure 2G,H). This *m*/*z* 570 value is exactly two units less than twice the molecular weight of scutellarein (M.W. 286). Thus, we initially assumed that two scutellarein molecules dimerized forming a covalent bond.

In the UPLC-ESI-Q-TOF-MS/MS analysis of scutellarin, the peak intensity was so weak that it could not produce MS or MS/MS spectra (Figure 2I,J). This means that the RAF potential of scutellarin is negligible. Hence, the difference in antioxidant capacities between scutellarein and scutellarin can be attributed to glucuronidation at 7-*O*-position, as shown in Figure 1. Such 7-*O*-glucuronidation not only reduces the amount of phenolic -OH, but also transforms the pyrogallol-type phytophenol into a catechol-type phytophenol. Given that both pyrogallol-type and catechol-type molecules undergo HAT to reveal their antioxidant nature, their antioxidant levels rely on the BDE (bond dissociation enthalpy or bond dissociation energy) value of the first phenolic -OH. However, physical chemistry calculations based on the B3LYP method revealed that the BDE value for pyrogallol-type molecules (289.4 kJ/mol) is lower than that for catechol-type molecules (312.8 kJ/mol) [37]. Thus, 7-*O*-glucuronidation increased the BDE value and subsequently decreased the antioxidant capacity. Moreover, because 7-*O*-glucuronidation has already occupied the covalent bonding site, DPPH^•^ can hardly be connected to form relevant RAF products.

Now, it is clear that glucuronidation plays a detrimental role by reducing the antioxidant capacities of phytophenols. This finding can also explain the difference in antioxidant capacities between other pairs of phytophenols and related metabolites, including ferulic acid and its metabolite ferulic acid-4′-*O*-glucuronide [22], quercetin and its metabolite quercetin-3-*O*-glucuronide [38], silymarin and its metabolite silibinin-glucuronide [4], and (-)-epicatechin and its metabolite (-)-epicatechin-glucuronide [39].

Contrary to the above antioxidant analyses, ^•^O_2_^−^-scavenging analysis indicated that scutellarein was inferior to scutellarin (Appendix A and Table 1). This is because scutellarein maintains the pyrogallol-type structure, which has been demonstrated to produce ^•^O_2_^−^ anions under pH 7.4 [40]. Thus, the glucuronidation of one phenolic -OH in pyrogallol-type molecules can promote ^•^O_2_^−^ scavenging.

In addition, in Fe^2+^-binding UV-Vis analysis, scutellarein was found to be inferior to scutellarin. As illustrated in Figure 4A, after mixing with Fe^2+^, scutellarin displayed a greater change in the intensity of the UV peak than scutellarein, despite the fact that both scutellarein and scutellarin were observed to produce two red-shifts in the UV spectra (284 nm→300 nm and 338 nm→367 nm for scutellarein and 284 nm→296 nm and 338 nm→347 nm for scutellarin). In the visible spectra (Figure 4B), scutellarin yielded a visible peak at 611 nm; by comparison, scutellarein could hardly produce a similar visible peak. Such difference between scutellarein and scutellarin is certainly due to glucuronidation, which introduced a β-d-glucopyranuronic acid moiety into the scutellarin molecule. As shown in the preferred conformation (Figure 5), the β*-*d-glucopyranuronic acid moiety is near the B-ring in the scutellarin molecule, which provides it with the possibility to participate in the Fe^2+^-binding reaction of the B-ring. Based on previous studies [41,42,43,44,45], their Fe^2+^-binding reaction can be proposed as in Figure 6.

Thus, the β-d-glucopyranuronic acid moiety might devote its carboxyl *O*-atom to binding to Fe^2+^ from the top side of the plane of the ring (Figure 6). Such binding reaction can actually be regarded as a chelating reaction, similar to ethylene diamine tetraacetic acid (EDTA) chelation. Scutellarein, however, does not have this Fe^2+^-chelating ability, as it lacks β-d-glucopyranuronic acid moiety. Thus, scutellarin showed a higher Fe^2+^-binding potential than scutellarein.

Fe^2+^ can promote ROS generation via the Fenton reaction (H_2_O_2_ + Fe^2+^→^•^OH + OH^−^ + Fe^3+^) or the Haber-Weiss reaction (Fe^3+^ + ^•^O_2_^−^→Fe^2+^ + O_2_
*plus*
^•^O_2_^−^ + H_2_O_2_→^•^OH + OH^−^ + O_2_). Thereby, Fe^2+^ chelation can effectively scavenge ROS and thus, is termed an indirect antioxidant mechanism. Now it is clear that the ^•^O_2_^−^-scavenging potential and the indirect antioxidant potential (i.e., Fe^2+^-binding) of scutellarin are superior to those of scutellarein, and glucuronidation plays a beneficial role in enhancing the antioxidant capacities of pyrogallol-type phytophenols.

## 3. Materials and Methods

### 3.1. Chemicals

Scutellarein (C_18_H_10_O_6_, CAS number: 529-53-5, M.W 286.2, purity 98%) and scutellarin (C_21_H_18_O_12_, CAS number: 27740-01-8, M.W 462.4, purity 98%) were obtained from BioBioPha Co., Ltd. (Kunming, China). The molecule 1,1-Diphenyl-2-picrylhydrazyl radical (DPPH^•^, C_18_H_12_N_5_O_6_) was obtained from Aladdin Chemical, Ltd. (Shanghai, China), (NH_4_)_2_ABTS [2,2′-azino-bis (3-ethylbenzo-thiazoline-6-sulfonic acid diammonium salt)] was obtained from Amresco Chemical Co (Solon, OH, USA), 2-phenyl-4,4,5,5-tetramethylimidazoline-1-oxyl-3-oxide radical (PTIO^•^) was purchased from TCI Chemical Co. (Shanghai, China), 2,9-Dimethyl-1,10-phenanthroline (neocuproine), pyrogallol, and (±)-6-hydroxyl-2,5,7,8-tetramethlychromane-2-carboxylic acid (Trolox) were obtained from Sigma-Aldrich (Shanghai, China). Tris-hydroxymethyl amino methane (Tris) was obtained from Dingguo Biotechnology, Ltd. (Beijing, China). Water and acetonitrile were of HPLC grade. Analytical-grade FeCl_3_·6H_2_O and other reagents were purchased from Guangdong Guanghua Chemical Plants Co., Ltd. (Shantou, China).

### 3.2. DPPH^•^-Scavenging Spectrophotometric Analysis

DPPH^•^ radical scavenging was conducted following a previously reported procedure [46]. The experimental protocols, experimental apparatus, and formula for calculating the inhibition percentages were similar to those previously reported. In contrast to the previous report, the samples tested in this study were scutellarein and scutellarin, with Trolox used as the positive control. The final concentrations of scutellarein and scutellarin were shown in Appendix A.

### 3.3. ABTS^+•^-Scavenging Spectrophotometric Analysis

ABTS^+*•*^ scavenging activity was evaluated by a reported method [47]. ABTS^+*•*^ was produced by mixing 350 μL of aqueous ABTS diammonium salt solution (7.4 mM) with 350 μL of aqueous potassium persulfate (2.6 mM). The mixture was kept in the dark at room temperature for 12 h to complete the radical generation and then diluted with methanol (about 1:50) so that its absorbance at 734 nm was 0.30 ± 0.02. To determine the scavenging activity, the test sample (*x* = 0–10 μL, 0.1 mg/mL) was added to (20 − *x*) μL of methanol followed by 80 μL of ABTS^+*•*^ reagent, and the absorbance at 734 nm was measured 2 min after the initial mixing, using methanol as the blank. The ABTS^+*•*^ inhibition percentage was calculated as follows:(1)$$Inhibition\;\%=\frac{A_{0}-A}{A_{0}}\times 100\%$$
where *A*_0_ is the absorbance of the control without the sample, and *A* is the absorbance of the reaction mixture with the sample.

### 3.4. PTIO^•^-Scavenging Spectrophotometric Analysis

The PTIO*^•^*-scavenging spectrophotometric assay was conducted following the method mentioned in our earlier work [28]. In brief, a dimethyl sulfoxide (DMSO) solution of a test sample (*x* = 0–10 μL, 1.0 mg/mL) was added to (20 − *x*) μL of methanol, followed by the addition of 80 μL of an aqueous PTIO*^•^* solution. The aqueous PTIO*^•^* solution was prepared using a 0.1 mM phosphate-buffer solution (pH 7.4). The mixture was maintained at 37 °C for 1 h, and the absorbance was then measured at 560 nm using a microplate reader (Multiskan FC, Thermo scientific, Shanghai, China). The phosphate-buffer solution was used as a blank. The percentage inhibition was calculated on the basis of the formula presented in Section 3.3.

### 3.5. Cu^2+^-Reducing Antioxidant Spectrophotometric Analysis

The cupric ion-reducing antioxidant capacity (CUPRAC) assay was performed on the basis of the method proposed by Apak et al. [48], with small modifications as proposed by Jiang [42]. A volume of 12 μL of aqueous CuSO_4_ solution (0.01 M) and 12 μL of methanol neocuproine solution (7.5 × 10^−3^ M) was added into the wells of a 96-well plate and mixed with different concentrations of the samples (4–20 μg/mL). The total volume was then adjusted to 100 μL with a CH_3_COONH_4_ buffer solution (0.1 M) and mixed again to homogenize the solution. The mixture was maintained at room temperature for 45 min, and the absorbance was measured at 450 nm on a microplate reader against the buffer (as a blank). The relative reducing power of the sample was calculated using the formula:(2)$$Relative\;reducing\;effect\;\%=\frac{A-A_{\min}}{A_{\max}-A_{\min}}\times 100\%$$
where *A* is the absorbance of the sample at 450 nm, *A*_max_ is the maximum absorbance at 450 nm, and *A*_min_ is the minimum absorbance at 450 nm.

### 3.6. Superoxide Anion (^•^O_2_^−^)-Scavenging Spectrophotometric Analysis (Pyrogallol Autoxidation Method)

The superoxide anion (^•^O_2_^−^)-scavenging activity was determined using a method previously developed in our laboratory [40]. Briefly, 10–50 μL of DMSO sample solution (0.5 mg/mL) was added to 0.05 M Tris-HCl buffer (pH 7.4) containing Na_2_EDTA (1 mM), and the total volume was adjusted to 980 μL using the buffer solution. Then, 20 μL of pyrogallol (1,2,3-trihydroxylbenzene) solution (60 mM in 1 mM HCl) was added to the sample, and the resulting mixture was vigorously agitated before being analyzed at 325 nm in 30 s intervals for 5 min. The ^•^O_2_^−^ radical-scavenging ability was calculated as follows:(3)$$Inhibition\;\%=\;\frac{\left(\frac{\Delta A_{325\;nm,control}}{T}\right)\;-\;\left(\frac{\Delta A_{325\;nm,sample}}{T}\right)\;}{\left(\frac{\Delta A_{325nm,control}}{T}\right)\;}\times 100\%$$
where Δ*A_325 nm, control_* and Δ*A_325 nm, sample_* are the increase in the *A_325 nm_* values of the mixture without and with the sample, respectively, and *T* is the time required for the determination (5 min in this case).

### 3.7. UPLC-ESI-Q-TOF-MS/MS Analysis of DPPH^•^ Reaction Products with Scutellarein (or Scutellarin)

Scutellarein (or scutellarin) was reacted with DPPH^•^ under the conditions described in earlier literature [49]. In brief, a solution of scutellarein (or scutellarin) in methanol was mixed with a solution of DPPH^•^ in methanol in a molar ratio of 1:2, and the resulting solution was incubated for 10 h at room temperature. The product was then filtered through a 0.22 μm filter and used for UPLC-ESI-Q-TOF-MS/MS analysis.

UPLC-ESI-Q-TOF-MS/MS analysis was carried out following the method mentioned in our previous work [50]. The chromatographic system was equipped with a C_18_ column (2.0 mm i.d. × 100 mm, 2.2 μm, Shimadzu Co., Kyoto, Japan). The mobile phase used for the elution of the system consisted of a mixture of acetonitrile (phase A) and 0.1% formic acid in water (phase B). The column was eluted at a flow rate of 0.2 mL/min with the following gradient elution program: 0–2 min, 30% B; 2–10 min, 30%–0% B; 10–12 min, 0%–30% B. The sample injection volume was set at 1 μL for the separation of the different components. The Q-TOF-MS/MS system was equipped with a Triple TOF 5600*^plus^* Mass spectrometer (AB SCIEX, Framingham, MA, USA) with an ESI source. The scan range was set to 100–2000 Da in the negative ionization mode. The system was run with the following parameters: ion spray voltage, −4500 V; ion source heater, 550 °C; curtain gas (CUR, N_2_), 30 psi; nebulizing gas (GS1, Air), 50 psi; Tis gas (GS2, Air), 50 psi. The declustering potential (DP) was set to −100 V, whereas the collision energy (CE) was set to −40 V with a collision energy spread (CES) of 20 V. The final RAF products were quantified by extracting the corresponding molecular formulae (e.g., [C_33_H_21_N_5_O_12_-H]^−^ for scutellarein–DPPH•; [C_30_H_18_O_12_-H]^−^ for the scutellarein-scutellarein dimer) from the total ion chromatogram and integrating the corresponding peak using PeakView 2.0 software (AB Sciex, Framingham, MA, USA).

The corresponding molecular formulae extracted in the case of scutellarin were [C_39_H_29_N_5_O_18_-H]^−^ for scutellarin–DPPH^•^ and [C_42_H_34_O_24_-H]^−^ for the scutellarin-scutellarin dimer.

### 3.8. Fe^2+^-Chelating Assay by Ultraviolet-Visible (UV-Vis) Spectra Analysis

The Fe^2+^-chelating ability was assessed by UV-Vis spectroscopy [41]. In brief, 100 μL of sample in methanol (0.7 mmol/L) and 100 μL of aqueous FeCl_2_·4H_2_O solution (7.04 mmol/L) were added to 300 μL of methanol/water (1:1, *v*/*v*) and mixed well. The resulting mixture was subsequently scanned using a UV-Vis spectrophotometer (Unico 2600A, Shanghai, China) from 200 to 900 nm within an hour. Methanol/water (1:1, *v*/*v*) served as a blank. Next, 200 μL of the mixture was transferred into a 96-well plate and photographed using a camera.

### 3.9. Statistical Analysis

The results were reported as the mean ± SD values of three independent measurements. The IC_50_ values were calculated by linear regression analysis. Independent-sample *t*-tests were performed to compare different groups. A *p* value of less than 0.05 was considered statistically significant. Statistical analyses were performed using the SPSS software 17.0 (SPSS Inc., Chicago, IL, USA) for windows. All linear regression analyses described in this paper were processed using version 6.0 of the Origin professional software.

## 4. Conclusions

Glucuronidation has a dual effect on pyrogallol-type phytophenol antioxidants: On the one hand, it reduces the antioxidant potentials (except for ^•^O_2_^−^-scavenging potential), on the other hand, it enhances the ^•^O_2_^−^-scavenging and Fe^2+^-binding potentials.

## Figures and Tables

**Figure 1 molecules-23-03225-f001:**
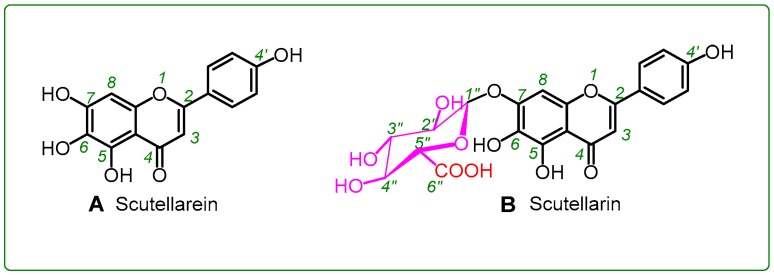
Structures of scutellarein (**A**) and scutellarin (i.e., scutellarein-7-*O*-glucuronide, (**B**)).

**Figure 2 molecules-23-03225-f002:**
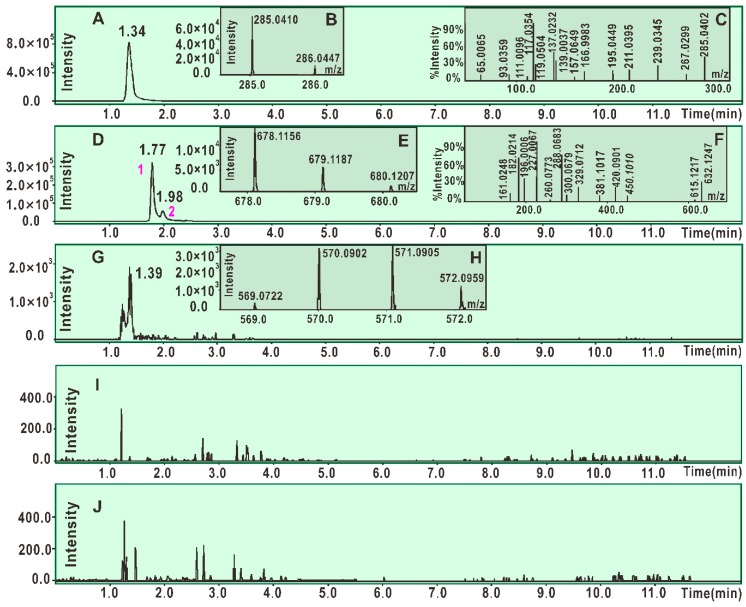
Main results of UPLC-ESI-Q-TOF-MS/MS analysis. (**A**) Chromatogram of scutellarein when the formula [C_15_H_10_O_6-_H]^−^ was extracted; (**B**) primary MS spectra of scutellarein. (**C**) Secondary MS spectra of scutellarein; (**D**) chromatogram of the radical adduct formation (RAF) product, scutellarein-DPPH, when the formula [C_33_H_21_N_5_O_12_-H]^−^ was extracted; (**E**) primary MS spectra of the RAF product scutellarein-DPPH (from peak 1 and peak 2); (**F**) secondary MS spectra of the RAF product scutellarein-DPPH (from peak 1); (**G**) chromatogram of possible dimeric products of scutellarein when the formula [C_30_H_18_O_12_-H]^−^ was extracted; (**H**) primary MS spectra of possible dimeric products of scutellarein; (**I**) chromatogram of the RAF product scutellarin-DPPH when the formula [C_39_H_29_N_5_O_18-_H]^−^ was extracted; (**J**) chromatogram of possible dimeric products of scutellarin when the formula [C_42_H_34_O_24_-H]^−^ was extracted.

**Figure 3 molecules-23-03225-f003:**
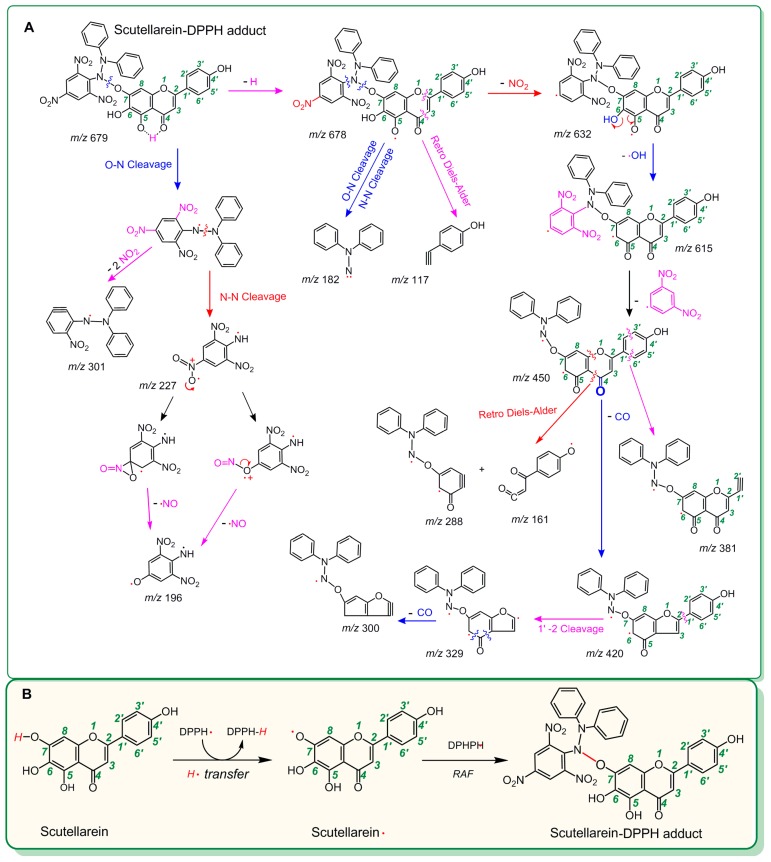
MS elucidation of the scutellarein-DPPH adduct structure (**A**) and the proposed reaction for the formation of the scutellarein-DPPH adduct (**B**). (The MS spectra are in negative ion mode, and the charge imposed by the MS field is not marked. Other linking sites between the scutellarein moieties and the DPPH moiety should not be excluded; other reasonable cleavages should not be excluded in the MS elucidation.).

**Figure 4 molecules-23-03225-f004:**
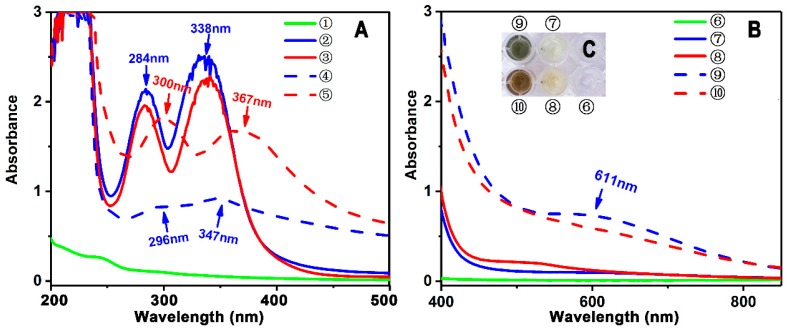
Experimental results of UV-vis spectra analysis of Fe^2+^-chelation with scutellarein and scutellarin. (**A**) Wavelength 200–500 nm; (**B**) wavelength 400–900 nm (① 0.90 mmol/L Fe^2+^; ② 0.09 mmol/L scutellarin; ③ 0.09 mmol/L scutellarein; ④ reaction mixture of 0.90 mmol/L Fe^2+^ with 0.09 mmol/L scutellarin; ⑤ reaction mixture of 0.90 mmol/L Fe^2+^ with 0.09 mmol/L scutellarein.); (⑥ 7.00 mmol/L Fe^2+^; ⑦ 0.70 mmol/L scutellarin; ⑧ 0.70 mmol/L scutellarein; ⑨ reaction mixture of 7.00 mmol/L Fe^2+^ with 0.70 mmol/L scutellarin; ⑩ reaction mixture of 7.00 mmol/L Fe^2+^ with 0.70 mmol/L scutellarein.); (**C**) appearance of the solutions.

**Figure 5 molecules-23-03225-f005:**
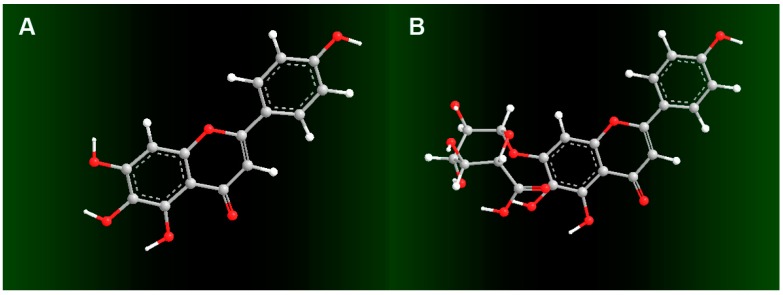
Preferred conformational ball-and-stick models of scutellarein (**A**) and scutellarin (**B**). The ball-and-stick models were created using Chem3D Pro 14.0.

**Figure 6 molecules-23-03225-f006:**
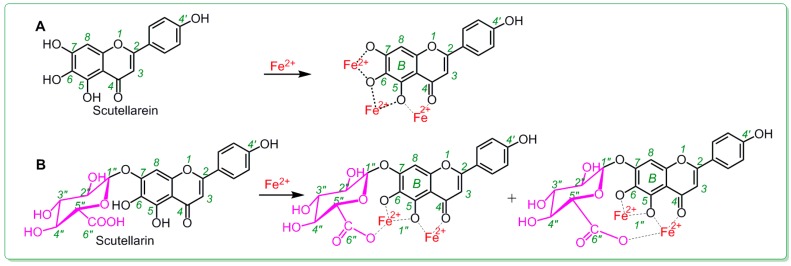
Proposed Fe^2+^-binding reactions of scutellarein (**A**) and scutellarin (**B**).

**Table 1 molecules-23-03225-t001:** IC_50_ values (μM) of scutellarein and scutellarin in five antioxidant spectrophotometric assays.

Antioxidant Assays	Scutellarein	Scutellarin	Trolox
DPPH^•^-scavenging assay	18.7 ± 0.1 ^a^	24.2 ± 1.7 ^b^	20.2 ± 0.5 ^a^
ABTS^+•^-scavenging assay	18.3 ± 1.2 ^a^	33.3 ± 2.9 ^c^	23.7 ± 0.4 ^b^
PTIO^•^-scavenging assay	177.5 ± 7.8 ^a^	577.2 ± 75.4 ^b^	185.7 ± 9.0 ^a^
Cu^2+^-reducing assay	33.5 ± 1.4 ^a^	43.4 ± 1.5 ^b^	61.5 ± 2.0 ^c^
^•^O_2_^−^-scavenging assay	79.0 ± 0.5 ^b^	28.8 ± 1.4 ^a^	291.5 ± 40.6 ^c^

The IC_50_ value (in μM units) is defined as the final concentration of 50% radical inhibition or relative reducing power, calculated by linear regression analysis and expressed as the mean ± SD (*n* = 3). Linear regression was analyzed by version 6.0 of the Origin professional software. The IC_50_ values with different superscripts (a, b, or c) in the same row are significantly different (*p* < 0.05). Trolox is the positive control. The dose–response curves are shown in Appendix A. DPPH^•^, 1,1-diphenyl-2-picrylhydrazyl, ABTS^+•^-, 2,2′-azino-bis (3-ethylbenzothiazoline-6-sulfonic acid), PTIO^•^, 2-phenyl-4,4,5,5-tetramethylimidazoline-1-oxyl 3-oxide radicals.

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
