# Peer review of "Dual Effect of Glucuronidation of a Pyrogallol-Type Phytophenol Antioxidant: A Comparison between Scutellarein and Scutellarin"

_molecules, 2018, doi:10.3390/molecules23123225_

Round 1
Reviewer 1 Report
The manuscript is interesting, well-structured in general and based on clearly described experiments. The Authors performed number of assays to reveal that the glucuronidation of pyrogallol-type phytophenol antioxidant imposes a dual effect. Moreover, these findings could be important for a reasonable number of scientists. I recommend publication of this study in its present form.
Author Response
Thank you very much!
Reviewer 2 Report
The manuscript is very interesting and well written. However, major revision is required to make it in a form acceptable for publication.
1) As regards UV-Vis spectra, authors state that they registered between 200 and 900 nm. However, they show them only partially. They should show the entire spectra, since it is also of importance the UV region, especially for comparison between the free ligands and the Fe(II)-complexes.
2) Authors should explain in more details the procedure in respect to the blank used in the UV-Vis spectra registration. Did they use one blank only? Or there was a different blank for each sample?
3) Authors should show more comparable spectra, regarding the maximum absorbance in the UV-Vis spectra.
4) At pag 9, line 311, authors use the term "supernatant". What exactly does that mean? Is there any formation of precipitate? Moreover, the brown colour of the picture of sample 5 in Figure 4 could point towards the formation of oxidized compounds.
5) To support the hypothesis of the involvement of the COOH group of the glucuronic moiety, it could be important to test the behaviour of scutellarein-7-O-glucoside, since this latter compound would be expected not to behave as scutellarein. Moreover, scutellarein-7-O-glucoside has no pyrogallol-type moiety and should therefore be able to scavenge •O2- as well.
6) Authors should differenziate among results in which one compound is comparable with the standard (DPPH, ABTS, PTIO), and others, in which, both compounds reveal a major effect than Trolox. In the case of Cu2+-reducing power, also scutellarin is more effective than Trolox due to the cathecol moiety.
Minor:
1) Authors should introduce abbreviations in the abstract for ABTS, PTIO, as they do for DPPH
2) Authors should check the use of superscripts in Figure S1, since they use only "a" and "c" and not "b"
3) Pag 4, line 129: please change "this" with "these"
Reviewer 3 Report
This is a reasonable and interesting reaserch paper. My comments
a) Do we have a Fe(III) vs. substrate study?
b) The spectrophotometric measurements are actually visible (not UV-visible)
c) Why not use structure confirmation with NMR?
d) Excellent introduction
e) I wish I could see some chemical equations (line 53, lines 65-67, lines 95-97).
f) Question- and please forgive my ignorance: acetals are easily hydrolyzed in aqueous acid. So would the acetals be hydrolyzed in the stomach and the antioxiant activity increase?
g) Fig 5 is exceptional- congratulations!
h) line 183: it should read "can be proposed"
i) line 202: is should read "via the Fenton reaction"
j) provide equations for line 202.
k) excellent decrription of the chemicals' list. I never knew one could get a bottle of radical such as PTIO'
l) Outstanding list of references- did you think about writing a review?
Round 2
Reviewer 2 Report
Authors did improve their manuscript according to most of the comments and suggestions.
In particular, in response to my comment #5, they tested a new compound (Oroxin A), to support their hypothesis of the role of COOH.
However, there is no trace of the results obtained neither in Table S4 and Figure S5, nor in the Discussion.
Therefore, authors are strongly suggested to insert these data and modify Discussion accordingly.
Author Response
Dear reviewer,
The results and structure of oroxin A have been included in the revised Suppl. 1 (p. 5).
However, it is not suitable to insert these results into main text. This is because that, (1) The sole structural difference beween scutellarein and i scutellarin is glucuronidation of 7-OH, thus, the antioxidant difference can only be attributed to the glucuronidation of 7-OH. The deduction is logical. (2) The addition of oroxin A may impair the completeness of main text.
Thank you very much.
Xican Li
